# Triaxial Carbon Nanotube/Conducting Polymer Wet-Spun Fibers Supercapacitors for Wearable Electronics

**DOI:** 10.3390/nano11010003

**Published:** 2020-12-22

**Authors:** Azadeh Mirabedini, Zan Lu, Saber Mostafavian, Javad Foroughi

**Affiliations:** 1Faculty of Science, Engineering and Technology, Swinburne University of Technology, Melbourne, VIC 3122, Australia; azade.mira@gmail.com; 2School of Textiles and Fashion, Shanghai University of Engineering Science, Shanghai 201620, China; luzandhu@gmail.com; 3Intelligent Polymer Research Institute, AIIM Facility, University of Wollongong, North Wollongong, NSW 2500, Australia; Saber_mostafavian@yahoo.com; 4School of Electrical, Computer and Telecommunications Engineering, Faculty of Engineering and Information Sciences, University of Wollongong, Keiraville, NSW 2522, Australia; 5Westgerman Heart and Vascular Center, University of Duisburg-Essen, 45122 Essen, Germany

**Keywords:** fiber supercapacitor, wet-spinning, e-textile, wearable electronics

## Abstract

The ubiquity of wearables, coupled with the increasing demand for power, presents a unique opportunity for nanostructured fiber-based mobile energy storage systems. When designing wearable electronic textiles, there is a need for mechanically flexible, low-cost and light-weight components. To meet this demand, we have developed an all-in-one fiber supercapacitor with a total thickness of less than 100 μm using a novel facile coaxial wet-spinning approach followed by a fiber wrapping step. The formed triaxial fiber nanostructure consisted of an inner poly(3,4-ethylenedioxythiophene) polystyrene sulfonate (PEDOT:PSS) core coated with an ionically conducting chitosan sheath, subsequently wrapped with a carbon nanotube (CNT) fiber. The resulting supercapacitor is highly flexible, delivers a maximum energy density 5.83 Wh kg^−1^ and an extremely high power of 1399 W kg^−1^ along with remarkable cyclic stability and specific capacitance. This asymmetric all-in-one fiber supercapacitor may pave the way to a future generation of wearable energy storage devices.

## 1. Introduction

Within the last couple of years, an expansive effort has begun for development of ultrathin portable and flexible electronic devices and their integration into various wearable systems which has motivated the exploration for appropriate energy supply [1,2,3,4,5,6]. The incorporation of electronic components into common textile structures could facilitate free and easy access while allowing a number of smart functionalities such as sensing, actuating, energy storage or information processing [4,7]. In addition, fabrics provide an efficient basis to work with since they can be easily shaped into the human body form as well as providing easy access to the electronic equipment embedded within [8]. Despite the advances in wearable electronic elements, there is still a need to develop high-performance, flexible energy-storage devices, with adequate flexibility and robustness.

Supercapacitors (or ultracapacitors), that possess higher energy density than traditional dielectric capacitors and deliver much higher power rates than batteries, are known to be at the leading edge of electrical energy storage research [9]. In the past few decades, planar supercapacitors (SCs) have been extensively studied and used [9,10,11,12,13,14] mainly due to their exceptional electrical properties such as fast charge and discharge rates, high power density, safety and long lifetime [15,16]. Apart from their light weight and flexibility, they need further improvements of their energy storage and power delivery to enable use as wearable devices [17,18]. Fiber-based SCs with diameters of about tens to hundreds of micrometers have lately been shown to be promising candidates as energy storage structures for wearable energy storage devices [15,19,20,21]. Compared to conventional supercapacitive devices, fibrillar SCs are designed to meet the requirements of flexibility, foldability, lightness and conformability [17]. Those presented features make them ideal candidates since they can be directly embedded as components into textiles using facile knitting/braiding techniques.

Wet-spinning has been shown to be one of the most promising strategies for the formation of tubular SCs [9,22,23,24]. Formed wet-spun fibers exhibit suitable flexibility and mechanical properties while providing versatility in choice of materials and structures. The ability to be woven or knitted into fabric allows for wet-spun fibers to be utilized as SCs in wearable electronics. Recently, considerable effort has been devoted to developing so-called all-in-one fiber SCs wherein electrodes and solid electrolytes (as separators) have been integrated into a single entity.

In order to achieve an all-in-one SC device, a facile coaxial wet-spinning technique was employed allowing for the continuous production of coaxial fibers having a conductive core with a hydrogel sheath acting as the electrolyte. Coaxial wet-spinning involves two polymer feed solutions being injected into a coaxial spinneret and co-extruded into a bath while retaining a coaxial structure [25,26]. The second conductive layer was then incorporated using a facile conductive wrapping method whereby as-prepared core-sheath fibers (insulating sheath) will be covered with CNT layer. High ionic conductivity and appropriate mechanical properties are crucial factors in the selection of a solid-state electrolyte [27,28,29].

Poly(3,4-ethylenedioxythiophene) polystyrene sulfonate (PEDOT:PSS) is considered a promising material for SC electrodes mainly due to its excellent intrinsic properties such as high conductivity, high-redox active capacitance, good chemical and electrochemical stability and ease of processability [30,31]. Among the most significant of these properties is water solubility and ease of use in wet-spinning which facilitated the fabrication of fiber SCs [32,33] Chitosan is a cationic polysaccharide derived from crustacean skeletons and has previously been used to form gels suitable for use as a solid electrolyte with an ionic conductivity of the order of 10^−3^–10^−4^ Scm^−1^ [25,34]. In an acidic medium, chitosan amino groups are positively-charged and can thus react with the “free” negatively-charged polystyrene sulfonate acid (PSS) groups of PEDOT:PSS, according to the polyionic complexation coagulation strategy [35], to create hydrogen bonding at the interface between the core and the sheath in a coaxial fiber [36]. CNT fibers have been used previously as electrode material in SCs due to the combination of mechanical and electrical properties attainable [17,37,38,39,40,41]. Hybrid electrodes including both CNT and conducting polymer (CP) can take benefit from the large pseudocapacitance of the CPs coupled with the conductivity and mechanical strength of the CNT [7,40,42,43].

Researchers have investigated several strategies for the preparation of similar multi-ply fiber SCs [31,44,45,46,47,48,49,50,51,52,53,54]. Lee et al. described a biscrolled yarn-like SC based on CNT/PEDOT:PSS nanocomposite and a Pt wire with gel H_2_SO_4_/polyvinyl alcohol (PVA) polymer electrolyte, which indicated a specific capacitance of 0.46 mF cm^−1^ and excellent cycling performance under different mechanical modes [50]. In 2014, a two-ply gamma-irradiated CNT (IR-CNT)/PEDOT:PSS yarn SC was reported using PVA-H_3_PO_4_ gel electrolyte, resulting in maximum capacitance of 18.5 F g^−1^ with no significant reduction in capacitance after 600 charge-discharge cycles. A multi-step wet-spinning process was lately employed to produce ternary CNT/MnO_2_/PEDOT:PSS composite fiber SCs [44]. The assembled SC device exhibited a high specific capacitance of 51.3 F g^−1^ and good cycling stability of ~84% capacitance retention after 1000 cycles. A summary of reported hybrid similar fiber SCs is presented in Table 1.

With the final aim of developing compact, thin fibers as all-in-one tubular SCs, a facile coaxial wet-spinning approach was employed followed by a wrapping technique. Using this approach, long fiber SCs are achievable, where all the elements are integrated into one unified structure. Once produced, these fabricated fiber SCs may be easily incorporated into a textile form using knitting or braiding techniques.

## 2. Experimental

### 2.1. Materials

Chitosan (medium molecular weight, the degree of deacetylation ~80%) was obtained from Sigma Aldrich, Australia. Acetic acid was supplied from Ajax Finechem, Australia and used directly without further purification. Sodium hydroxide pellets (obtained from Ajax Finechem, Australia) were used as the coagulating agent. Isopropanol (ISP) was obtained from Merck Chemicals, Australia. PEDOT:PSS pellets were purchased from Agfa, Taiwan (Orgacon dry, Lot A6 0000 AC), and polyethylene glycol (PEG) with a molecular weight of 2000 ${g\;\mathrm{mol}}^{-1}$ was purchased from Fluka, Philadelphia, PA, USA. Sterile filters of 1 µm were supplied from EASYstrainer, Germany and used later to filter both spinning solutions before wet-spinning. MWNT forests were synthesized in-house, as previously discussed [21]. Sodium and lithium chloride salts, were also sourced from Sigma-Aldrich, Australia.

### 2.2. Preparation of Spinning Solutions

For the purpose of coaxial spinning, a chitosan solution was prepared such that the final concentration was 3% w v^−1^, as described in our previous work [25]. Furthermore, 2.5% w v^−1^ aqueous dispersions were made from PEDOT:PSS pellets. The dispersions were then homogenized (Labtek IKAR T25) at 18,000 rpm for 15 min, followed by 1 h of bath sonication (Branson B5500R-DTH). Finally, polyethylene glycol (PEG) (10% w v^−1^) was added as-prepared dispersion directly and further homogenized at 18,000 rpm for 2 min, followed by 10 min bath sonication.

### 2.3. Coaxial Wet-Spinning of Chitosan-PEDOT:PSS Fibers

The coaxial wet-spinning process was carried out as described before in our previous works [25,26]. PEDOT:PSS dispersions (with and without PEG) were injected through port B (Figure 1a) and extruded through the centre outlet nozzle into the appropriate coagulation bath. Simultaneously, chitosan was extruded as the sheath of the fiber, providing an outer casing for the core, by injection through port A (Figure 1), which facilitates extrusion through the outer segment of the spinneret nozzle.

Core-sheath fibers of chitosan-PEDOT:PSS (with or without PEG) (abbreviated as Chit-PEDOT for ease of use) were spun into a bath of 1 M aqueous NaOH (Ethanol/H_2_O: 1/5). The applied injection rate used for PEDOT:PSS dispersion was 15 mL h^−1^ and 28 mL h^−1^ for chitosan solution in order to provide sufficient time to cover the core material. Moreover, 28 mL h^−1^ was determined to be the optimum rate for the sheath component injection resulting in the thinnest sheath that was still capable of supporting the core material structure. Once the sheath components were coagulated, the fibers were then transferred into an isopropanol bath for post-treatment. After that, they were soaked in a graded series consisting of 80/20, 60/40, 40/60, 20/80 and 0/100 ISP/Milli-Q water mixtures gradually over 24 h as a washing step. To make an electrical connection to the conductive core embedded within core-sheath fibers, a cotton-steel wire with an average diameter of ~25 µm was inserted into the fiber core while spinning.

### 2.4. Fabrication of Triaxial CNT-Chitosan-PEDOT:PSS Fibers

For fabrication of triaxial CNT-Chit-PEDOT fibers, directly drawn nanotubes from a spinnable synthesized MWNT forest were wrapped around the surface of the as-prepared coaxial Chit-PEDOT fibers layer by layer (see Figure 1b), as previously reported [57]. After wrapping, drops of ethanol were applied to condense the MWNT layer and promote the interaction between the CNT and the fiber layers.

### 2.5. Characterizations

#### 2.5.1. Rheological Measurement

There are upper and lower practical limits which need to be considered in terms of suitable polymer concentrations for wet-spinning applications depending on the type of polymer as discussed in our previous works [25,26]. In addition to this, viscosity is regarded as the primary criterion for the selection of suitable concentrations of materials for the purpose of coaxial spinning. Thus, an understanding of the rheological properties of spinning solutions is essential to determine the optimum conditions required for the spinning process. Changes in viscosity were recorded as a function of shear rate between 0.1 and 300 s^−1^. The rheological properties of chitosan (3% w v^−1^ (with and without NaCl)) and PEDOT:PSS (3% w v^−1^) (with and without PEG) solutions were examined in flow mode (cone and plate method—60 mm diameter, 2° cone angle) using a digital rheometer- (AR G2, TA Instruments, New Castle, DE, USA).

#### 2.5.2. Impedance Behavior of Chitosan Hydrogel

The electrical impedance behavior of hydrogel samples (width = 5 mm, height = 6 mm) was measured using a custom-built impedance setup, described elsewhere [58]. In brief, an oscilloscope (Agilent U2701A) and waveform generator (Agilent U2761A) were used to apply a range of frequencies at a peak-to-peak ac voltage of 0.8 V, while measuring the voltage drop across a known resistor (10 kΩ) and the unknown hydrogel sample. The current was calculated across the known resistor and this was then used to calculate impedance behavior of the hydrogel samples as a function of frequency. Hydrogel samples tested varied in length (l) between 0.5 and 2.5 cm and were contacted at each end with reticulated vitreous carbon (RVC, ERG Aerospace, 20 pores per inch), as shown in Figure 2.

The RVC acted as a medium for contact between the rigid electrodes and soft gel. Electrical conductivity (*σ*) was calculated according to Equation (1) as below:(1)$$Z_{I}=\frac{l}{\sigma A_{C}}+R_{C}$$
where $Z_{I}$ is the frequency-independent impedance, $R_{C}$ is the contact resistance and $A_{C}$ is the cross-sectional area of the sample.

#### 2.5.3. Mechanical Analysis

Tensile experiments were performed with the aid of a dynamic mechanical analyzer (EZ-L, Shimadzu, Kyoto, Japan) with gauge length of 11.5 mm and at an extension rate of 2 mm min^−1^. Average values of tensile strength and maximum strain were determined.

#### 2.5.4. Microscopy

Digital images and measurements of prepared fibers (in their wet state) were obtained using a Leica M205 stereomicroscope, Australia in tandem with LAS software, version 4.4. The surface and cross-sectional morphology of coaxial fibers were also examined in wet-state using a JSM-6490LV scanning electron microscope (SEM), MA, USA using the SMile View software. Images were recorded in high-vacuum mode at 15 kV operating voltage and a spot size setting of 45. Samples were prepared for imaging by immersion in simulated body fluid (SBF) beforehand until they were fully swollen, and then short lengths (about 5 mm) were removed, drained of excess medium and inserted into holes (~1 mm diameter) which had been pre-drilled into a small brass block. The block containing the mounted fibers was then plunged into liquid nitrogen for about 45 s. and a liquid nitrogen-cooled razor blade was run across the surface of the block to fracture the fibers. The block was then quickly transferred to the LV-SEM for examination.

The morphological properties of the dried coaxial Chit-PEDOT and triaxial CNT-Chit-PEDOT fiber surfaces and cross-sections were observed using a JEOL JSM-7500 FESEM, Japan. Samples for imaging were prepared by cutting cross sections in liquid nitrogen using a scalpel blade. They were then coated (EDWARDS Auto 306) with a thin (10 nm) layer of Pt to aid with imaging and minimize beam heating effects. Cross-sections were analyzed at 25 kV accelerating voltage and a spot size setting of 12 under high vacuum.

#### 2.5.5. Cyclic Voltammetry

Cyclic voltammetry (CV) curves of an optimized asymmetric supercapacitor were collected over different potential windows from 0.8 to 1.4 V with scan rates ranging from 10 to 1000 mV s^−1^, respectively to evaluate the electrochemical (EC) properties of asymmetric supercapacitor fiber and 50 cycles were performed. To investigate the optimal potential window of asymmetric supercapacitor, the electrochemical performance of each electrode was first measured using an E-Corder 401 interface and a potentiostat (EDAQ) in a three-electrode cell with Ag/AgCl, Pt mesh and deoxygenated 0.1 M phosphate buffered saline (PBS) aqueous solution as the reference electrode, counter electrode and electrolyte, respectively. The cotton-steel yarn which has already inserted into the inner PEDOT:PSS core of coaxial fibers was used to make an electrical connection.

#### 2.5.6. Supercapacitor Performance Testing

Galvanostatic charge-discharge experiments of asymmetric supercapacitor were carried out at a current density of 0.5 A g^−1^ as well as different potential windows from 0.8 to 1.4 V by using a battery testing device (Neware Technology Ltd., China) in a two-electrode system in deoxygenated 0.1 M PBS aqueous solution. The cells were discharged galvanostatically to a cut-off cell voltage of 0.01 V.

## 3. Results and Discussions

### 3.1. Rheological Characterization of Fiber Spinning Solutions

Viscosity is a key factor determining the selection of dopes for wet-spinning. With coaxial spinning, as it is needed to provide a continuous protective coating over the inner core material, the viscosity of the sheath spinning solution is particularly critical. The core component material must also possess a certain minimum viscosity to allow continuous spinnability. In addition to these, the proximity of viscosities of the two components is also an essential consideration for coaxial spinning.

Under atmospheric pressure and constant temperature, the solution viscosity depends directly on the concentration of the dissolved polymer. Broad ranges of concentrations varying from 2–15% w v^−1^ have been reported as practical, spinnable concentrations for chitosan [25,59,60]. A concentration of 3% w v^−1^ has been selected here as it is amenable to wet-spinning and results in a gel with high ionic conductivity. A concentration of 2.5% w v^−1^ PEDOT:PSS dispersion has been shown to provide a “spinnable” dope, producing continuous fibers over several meters [61]. However, after wet-spinning of these dopes into the coagulation bath, the observation of coaxial fiber cross-sections under the scanning electron microscope has shown that the PEDOT:PSS core section has a non-integrated texture. In order to achieve uniform core, we have investigated the effect of adding PEG into the PEDOT:PSS spinning formulation. Adding PEG is also known to enhance the conductivity of PEDOT:PSS fibers [33]. We found that by adding 10% w v^−1^ PEG, the spinnability of the formulation was unaffected, thereby affording a one-step fiber production method. In addition to this, a uniform core texture could now be obtained routinely enclosed within the outer sheath component. Figure 3 shows changes in viscosity versus the shear rate at different shear rates between 0.1 and 500 s^−1^ for aqueous solutions of chitosan at 3% w v^−1^, Chit-NaCl 1% w v^−1^, PEDOT:PSS at 2.5% w v^−1^ before and after adding PEG.

A spinning solution composed of 3% w v^−1^ chitosan was found to have a viscosity of ~6.3 Pa·s. Increasing the shear rate, the viscosity decreased, an indication of shear-thinning behavior. The addition of a monovalent salt such as sodium chloride (1% w v^−1^) caused a slight decrease in the viscosity of the hydrogel due to electrostatic screening [62].

With increased shear rate, the viscosity of the PEDOT:PSS solution also decreased, again indicating a shear thinning behavior. The viscosity of the chitosan solution was higher than that observed for PEDOT:PSS dispersions at low shear rates. The viscosity of the 2.5% w v^−1^ PEDOT:PSS dispersion was 4.8 Pa·s, while the viscosity of PEDOT:PSS with PEG added was approximately 6.7 Pa·s at shear rates close to zero (Figure 3). The shear rates were calculated to be about ~94 s^−1^ for Chit and Chit-NaCl as well as ~62 s^−1^ for PEDOT:PSS (+PEG) solutions at the nozzle which resulted in a viscosity of ~2.4 Pa·s, ~1.39 Pa·s and ~0.95 Pa·s, respectively, during the spinning process (shown in Figure 3; inset).

### 3.2. Electrical Characterization of Hydrogels

The ionic conductivity of the chitosan gel was studied, and the effect of salt addition was investigated using an impedance testing protocol reported previously [58]. Figure 4a shows a typical series of Bode plots for chitosan 3% w v^−1^ aqueous hydrogel measured at different lengths (0.5, 1, 1.5, 2 and 2.5 cm). In all cases, the impedance magnitude (|*Z*|) decreases with increasing frequency and becomes independent of frequency above 1 kHz. This observation infers that the system is dominated by ionic, rather than electronic, charge carriers, as is expected. Figure 4b shows a linear relationship between the frequency-independent impedance ($Z_{I}$) and sample length; the slope of this plot was used to calculate conductivity according to Equation (1).

Table 2 shows the ionic conductivity of the chitosan hydrogel with varying NaCl salt contents (0, 0.5 and 1.0% w v^−1^). As expected, the conductivity increased with increasing the salt content. It can be seen that 1% w v^−1^ NaCl exhibited the highest conductivity with less variability (due to the smaller errors reported in Table 2 based on three time measurements).

### 3.3. Microscopic Investigation of As-Prepared Fibers

#### 3.3.1. Morphological Observation in Wet-State

Stereomicroscopic images of the fiber surface and cross-section were obtained (Figure 5a,b). The coaxial fibers showed linear uniform structures with the PEDOT:PSS (+PEG) core evenly loaded inside the chitosan sheath. The diameters of the Chit-PEDOT fibers were measured to be about 580 µm in which the core was about 340 µm. The cross-sectional images of the coaxial fibers obtained using LV-SEM, before and after the addition of PEG to the core component (Figure 5c,d), revealed the effect of PEG in creating the crack-free consistent central core component. A porous, permeable texture is also noticeable in the chitosan constituent, which looks as if it has reduced in size after insertion of PEG (Figure 5d).

Utilizing coaxial wet-spinning, where demonstrated that inherently conductive material (electrode) and electrically insulating material (separator) could be incorporated into the fiber configuration, has simultaneously eliminated the need for involving any assembly processes.

#### 3.3.2. Morphological Observation in Dry-State

Obtaining the right morphology is one of the most crucial steps in designing a multilayered structure. The internal and surface microstructure of dehydrated Chit-PEDOT (including and without PEG) and CNT-Chit-PEDOT fibers were imaged under SEM, and the results are shown in Figure 6a–f. As is well known, wet-spinning yields fibers of a generally round or bean-shaped cross-section [61]. Although hydrated coaxial Chit-PEDOT fibers indicated disk-shaped circular cross-sections (Figure 6d), it fully collapsed into non-regular structures after dehydration, as shown in Figure 6a. A clear boundary could be found between the two components (Figure 6b). Looking at the surface pattern of coaxial Chit-PEDOT fiber in Figure 6c showed that it displayed a non-uniform and irregular cross-section, while high magnification SEM images of CNT-Chit-PEDOT fiber surface (Figure 6f) revealed a relatively dense arrangement of CNTs on fiber surface which are in contact with each other; also some short CNTs can be seen in the transverse direction creating T-junctions with aligned CNTs. The cross-sectional shape of CNT-Chit-PEDOT triaxial fibers (Figure 6d) displayed a bean-shaped irregular cross-section similar to those shown in Figure 6a by Chit-PEDOT fiber with three distinct layers (Figure 6f) for which the CNT layer had a diameter of about 1–2 µm as repeated, elsewhere [63].

### 3.4. Mechanical Properties

Ultimate stresses (MPa), ultimate strains (%) and Young’s moduli (GPa) were measured for the Chit-PEDOT (with and without PEG) and CNT-Chit-PEDOT fibers. Cross-sectional areas were calculated as perfectly circular with diameters equal to the largest measured width of each irregular fiber. Average values of tensile strength and maximum strain were determined. The mechanical properties of as-prepared fibers are summarized in Table 3.

Results obtained from stress–strain curves for as-spun coaxial and triaxial fibers showed a significant increase in robustness compared to the results previously reported for PEDOT:PSS fibers [61]. Young’s moduli of fibers were calculated to be 3.9 ± 0.1, 5.2 ± 0.6 and 4.6 ± 0.3 GPa for Chit-PEDOT (without PEG), Chit-PEDOT (including PEG) and CNT-Chit-PEDOT fibers, respectively. The results indicate that the presence of a hydrogel sheath plays a major role in increasing Young’s modulus compared to the pristine PEDOT:PSS fiber. Coaxial fibers of Chit-PEDOT (without PEG) and Chit-PEDOT (including PEG) exhibited ~18% and ~57% enhancement in Young’s modulus compared to the PEDOT:PSS counterpart. After wrapping the CNT fibers, Young’s modulus decreased. The modulus of hybrid materials was found to be highly dependent on the interaction between the existing layers in the hybrid material given that the load distribution efficiency at the interphase is determined by the degree of adhesion between the components. Our results may be explained by the creation of a weaker interfacial adhesion between CNT segment and chitosan sheath in triaxial fibers compared to the bonding strength between PEDOT:PSS and chitosan in coaxial fibers. This limits the proper load transfer from the CNT to the other component and reduces the obtained modulus for CNT-Chit-PEDOT fibers.

Analysis of the mechanical data also indicated ultimate stress of 125.0 ± 7.1 MPa with 15.8 ± 1.2% strain, 94 ± 3.5 MPa with 8 ± 1.9% strain and 160.3 ± 6.8 MPa with 7.5 ± 0.9% strain had been obtained for solid PEDO:PSS [64], Chit-PEDOT (without PEG) and Chit-PEDOT (including PEG) fibers, respectively. In other words, there is a loss of ∼25% of tensile strain in hybrid Chit-PEDOT fibers compared to that of solid PEDOT:PSS fibers. This may be compensated for by the addition of PEG into the core of coaxial Chi-PEDOT fibers. This outcome can be explained by the creation of a strong homogenous interfacial connection between the core and sheath sections in the presence of PEG, which causes efficient stress transfer when strain is applied to a tensile specimen. Moreover, Chit-PEDOT (without PEG) fibers were shown to withstand less stress before breakage. Triaxial fibers of CNT-Chit-PEDOT were shown to have ultimate tensile stress of 134.8 ± 3.3 MPa with ~4.5 ± 0.4% elongation at breakage. The creation of multilayered fibers with different sections allowing different properties, as along with their dissimilar responses to the applied stress, maybe the reason behind this reduction.

### 3.5. Cyclic Voltammetry

The electrochemical behavior of triaxial CNT-Chit-PEDOT fiber-based SCs was investigated in 0.1 M PBS aqueous solution (Figure 7). To investigate the optimal potential window of asymmetric SC, the electrochemical performance of each electrode was measured individually in a three-electrode system in 0.1 M PBS.

As shown in Figure 7a, the PEG-doped PEDOT:PSS electrode presents a stable potential window toward oxidation and reduction between −0.2 and 0.4 V afterwards, while the CNT sheet covered electrode exhibits a stable potential in the range of −1.0 to 0 V, resulting in an extended potential window of 1.4 V for the whole assembled asymmetric SC. The near-rectangular shape of the CVs obtained from both active electrodes suggests that the overall internal resistance is low, owing to their high electroactivities. It is worth mentioning that the cotton-steel wire showed negligible electroactivity as tested.

In order to match the charge balance between the two electrodes, the loaded active materials should follow the relationship $q^{+}=q^{-}$, in which the stored charge by each electrode is determined by the specific capacitance (*C*), the potential range for the charge/discharge process (Δ*E*), and the mass loading of each electrode (m) using Equation (2) [15,16,65].
(2)q=C×ΔE×m

The specific capacitances calculated from the CV curves in Figure 7a shows a value of 2.6 F g^−1^ and 25.1 F g^−1^ of PEDOT:PSS and CNT electrode, respectively. On the basis of the calculation, the optimal mass ratio between the two electrodes should be:$$\frac{m_{p}}{m_{c}}\;=\;19.9$$

To keep the charges balanced on both electrodes, we can control the length of coaxial Chit-PEDOT fiber (0.07 mg/cm mass loading of PEDOT:PSS) and the amount of wrapped CNT sheets. The electrochemical properties of assembled asymmetric SC have been tested in 0.1 M PBS electrolyte to satisfy a further application in the biomedical field. CV curves and galvanostatic charge-discharge curves at different potential windows have been illustrated in Figure 7b,c with a scan rate of 50 ${\mathrm{mV}\;s}^{-1}$ and a current density of 0.5 ${A\;g}^{-1}$, respectively. The CV curves enjoyed a rectangular shape even when the potential window increased up to 1.4 V, while the charge-discharge curves showing an asymmetric triangular shape without any significant IR drop.

As seen in Figure 7a, PEG-doped PEDOT:PSS electrodes exhibited a well-defined EC behavior corresponding to oxidation (0.3 V) and reduction (−0.1 V) of the PEDOT backbone in the aqueous electrolyte. These redox values for the all-in-one asymmetric SC are at potentials higher than the oxidation state of the PEDOT:PSS fiber core. This provides the possibility of switching between oxidized and reduced states for fiber SC whilst the PEDOT:PSS layer remains in its oxidized (conducting) state.

The higher operating voltage can not only enhance the energy density but also reduce the number of capacitors in series to achieve expected output voltage [30]. To be further investigated, the CV curves in the extended potential window of 1.4 V is demonstrated in Figure 7d, exhibiting an almost rectangular shape even at a high scan rate of 500 mV s^−1^. The specific capacitance of the prepared asymmetric supercapacitor was calculated to be 21.4 F g^−1^ from galvanostatic charge-discharge curves using Equation (3);
(3)$$C=\frac{4I\Delta t}{m\Delta V}$$
where ∆*t* is the discharge time, *I* is the current applied to one electrode, m is the mass of two electrodes in the asymmetric supercapacitor, ΔV is the potential window of the discharging process. Moreover, the energy density (*E* = *CV*^2^/2; *E*, energy density; C, capacitance and V, operating voltage) and power density (*P = E/*Δ*t*; *E*, energy density; ∆*t*, the discharge time) of the assembled SC were determined to be 5.83 Wh kg^−1^ and 1399 W kg^−1^, respectively.

## 4. Conclusions

Using a novel coaxial wet-spinning approach followed by a facile wrapping technique we were able to fabricate a thin, flexible and lightweight asymmetric all-in-one fiber SC in an attempt to satisfy the requirements of high-performance wearable energy storage devices. The high dispersibility of PEDOT:PSS in water facilitated the preparation of well-dispersed and homogenous spinning formulations from which Chit-PEDOT fibers were continuously fabricated for the first time. A second thin conductive layer of CNT web was then used to cover the coaxial fiber surface through the wrapping. Integration of electrodes, solid electrolytes and separators into a single entity would simultaneously eliminate the need for involving a packaging process while the risk of creating a short circuit could be prohibited.

The morphological, mechanical and electrochemical properties of these fiber-based SCs are discussed. The addition of 10% w v^−1^ PEG into the PEDOT:PSS core was found to have a negligible effect on the spinnability of this formulation, whilst affording the consistent and crack-free spinning of a core component that could be maintained consistently without breaking into the sheath, as was confirmed in LV-SEM images. Sodium chloride with 1% w v^−1^ concentration was chosen as the desired salt solution to be added to the electrolyte layer, giving the highest ionic conductivity of 15.1 mS cm^−1^ in chitosan with lower standard deviations when compared to lithium chloride. Mechanical property results obtained from stress–strain curves for as-spun coaxial fibers showed a significant increase in robustness for them compared to results previously reported for PEDOT:PSS fibers by others. CNT-Chit-PEDOT fiber SCs were shown to have slightly decreased ultimate tensile stress values of ~134.8 MPa with ~4.5% ultimate strain, as a result of dissimilar responses to the applied stress from different sections of the fibers. Furthermore, the asymmetric all-in-one fiber SC showed reasonable electroactivities with total specific capacitances of 21.4 F g^−1^ and extremely high-power density of 1399 W kg^−1^. This combination of good electroactivity, remarkable Young’s modulus and yield stress combined with the high flexibility of these fibers imply that they hold great promise for applications in wearable energy storage devices in the near future.

## Figures and Tables

**Figure 1 nanomaterials-11-00003-f001:**
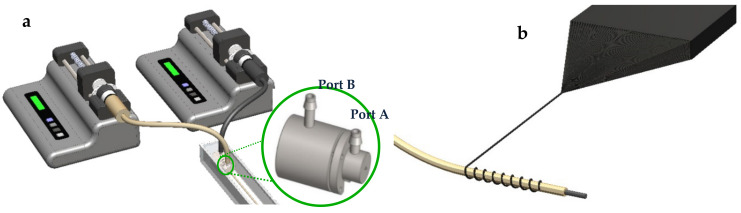
Schematic representation for (**a**) coaxial wet-spinning setup, and (**b**) CNT wrapping process.

**Figure 2 nanomaterials-11-00003-f002:**
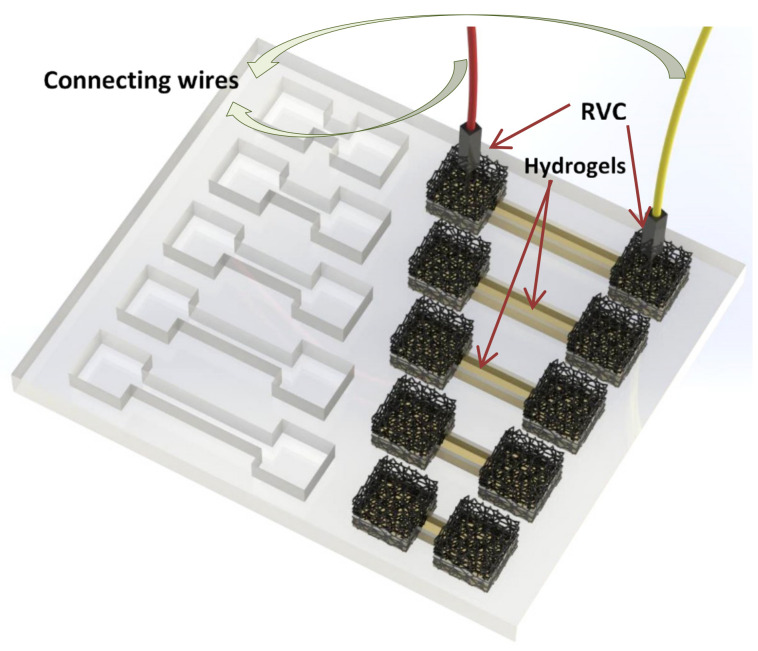
Hydrogel sample holder containing 5 channels varying in length between reticulated vitreous carbon (RVC) electrodes from 0.5 to 2.5 cm (height, 6 mm; channel width, 5 mm).

**Figure 3 nanomaterials-11-00003-f003:**
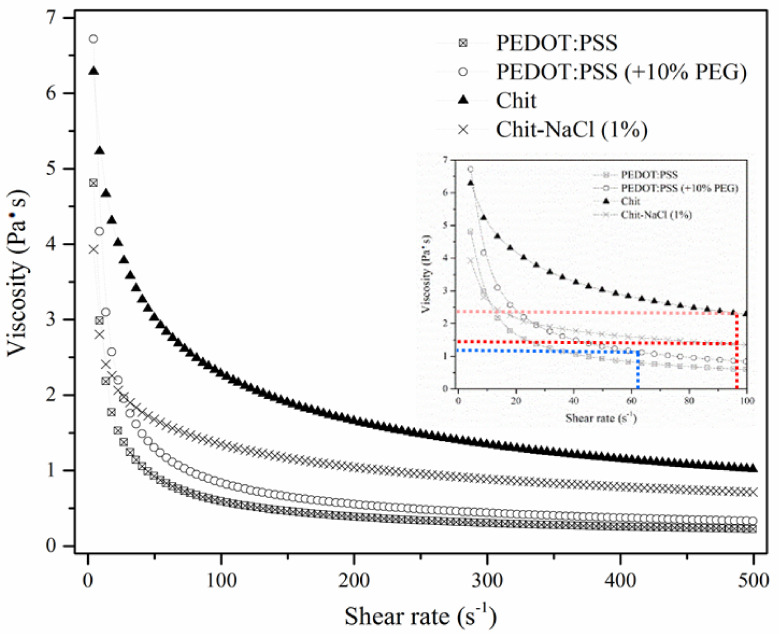
Viscosities of spinning solutions of PEDOT:PSS with and without addition of polyethylene glycol (PEG), chitosan and Chit-NaCl (1%) solutions, inset; viscosity changes vs. shear rate at different rates between 0.1 and 100 s^−1^.

**Figure 4 nanomaterials-11-00003-f004:**
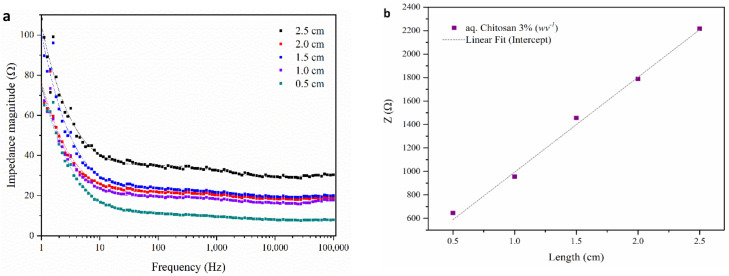
Electrical impedance analysis of chitosan hydrogel 3% w v^−1^ with 0.5% NaCl; (**a**) Bode plot and (**b**) impedance as a function of sample length in wet-state fibers.

**Figure 5 nanomaterials-11-00003-f005:**
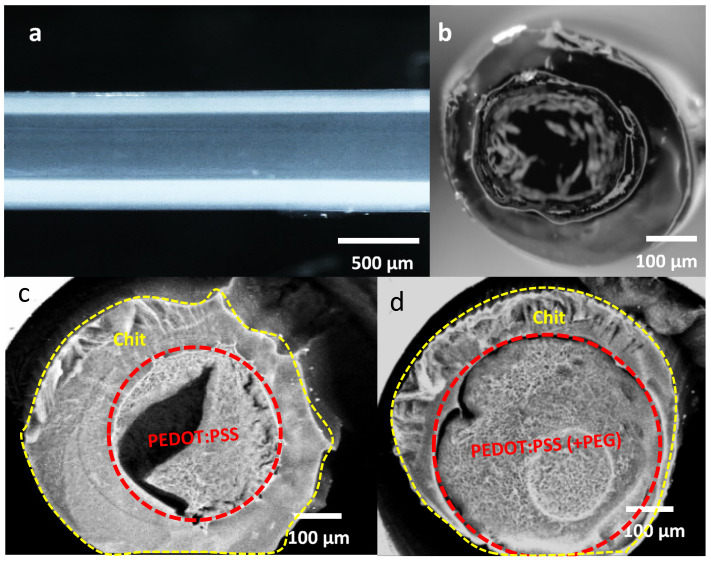
Stereomicroscope images of (**a**) surface and (**b**) cross-sections of Chit-PEDOT fibers, respectively; LV-SEM images of hydrated cross sections of as-prepared Chit-PEDOT fibers (**c**) without and (**d**) with addition of PEG.

**Figure 6 nanomaterials-11-00003-f006:**
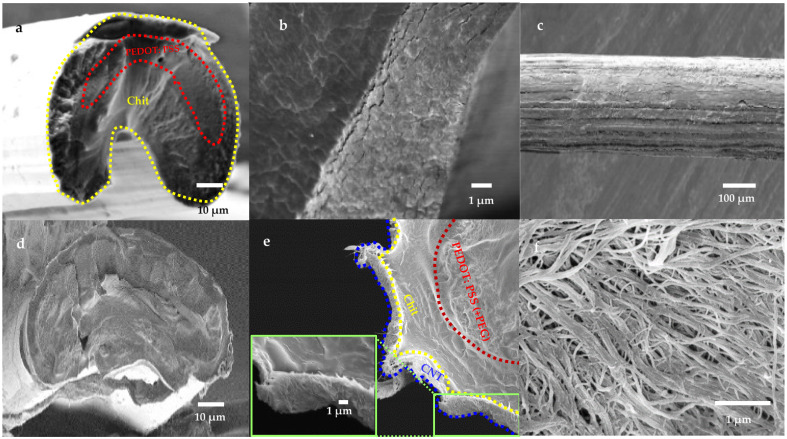
SEM images of (**a**,**d**) cross section, (**b**,**e**) higher magnifications and (**c**,**f**) surface pattern of as-prepared coaxial Chit-PEDOT and triaxial CNT-Chit-PEDOT fibers, respectively (the dashed lines are included to aid the reader in observing three different layers).

**Figure 7 nanomaterials-11-00003-f007:**
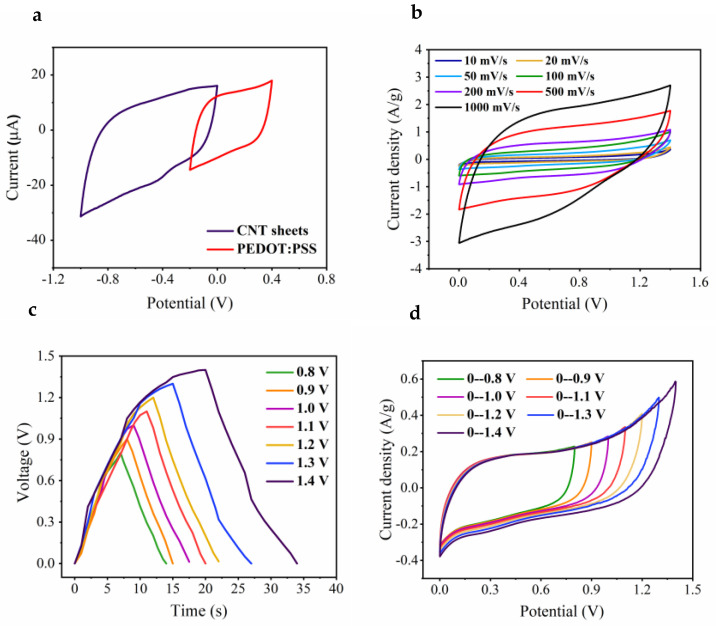
(**a**) Comparative cyclic voltammograms of PEDOT:PSS (+PEG) coated and CNT sheets wrapped cotton steel yarn electrodes obtained in a three-electrode system in 0.1 M PBS aqueous solution at a scan rate of 50 mV s^−1^. (**b**) CV curves tested over different voltages from 0.8 to 1.4 V at a scan rate of 50 mV s^−1^. (**c**) Galvanostatic charge-discharge curves obtained over different voltages from 0.8 to 1.4 V at a current density of 0.5 A g^−1^. (**d**) CV curves of an optimized asymmetric supercapacitor at scan rates ranging from 10 to 1000 mV s^−1^.

**Table 1 nanomaterials-11-00003-t001:** Summary performance of previously reported similar fiber supercapacitors (SCs).

Device Configuration	Electrolyte	Capacitance	Energy Density Power Density	Cycle Life	Ref.
IR-CNT/PEDOT:PSS	PVA/H_3_PO_4_	18.5 F g^−1^	—	91% (600 cycles)	[50]
MWNT^1^/PEDOT:PSS	PVA/H_2_SO_4_	0.46 mF cm^−1^	1.4 mWh cm^−3^ 40 W cm^−3^	92% (10,000 cycles)	[19]
PEDOT-coated polyester fabric	Na_2_SO_4_ solution	0.64 F cm^−2^ and 5.12 F cm^−3^			
PEDOT:PSS fiber	PVA/H_3_PO_4_	119 mF cm^−2^	4.13 µWh cm^−2^ 250 µW cm^−2^		[31]
Carbon cloth/PEDOT:PS S	PVA/H_3_PO_4_	73 F g^−1^ at 10 mV s^−1^	2.6 Wh kg^−1^ 2880 W kg^−1^	~100% (2000 cycles) ~95% under bending	[47]
DMSO ^2^ doped PEDOT:PSS coated onto cellulose/polyester cloth	Sweat (artificial and human)	8.94 F g^−1^ (10 mF cm^−2^) at 1 mV s^−1^	1.36 Wh kg^−1^ (1.63 µWh cm^−2^) at 1.31 V 329.70 W kg^−1^ (0.40 mW cm^−2^) at 1.31 V	87–45% (1000–5000 cycles)	[54]
MnO_2_/PEDOT:PSS/oxidized CNT fibers	H_2_SO_4_ aqueous solution	278.6 mF cm^−2^	125.37 μWh cm^−2^	92% (3000 cycles)	[55]
Carbon/MnO_2_/PEDOT:PSS composite fibres	PVA/H_3_PO_4_	51.3 F g^−1^	—	84.2% (1000 cycles)	[44]
MnO_2_/PEDOT:PSS/oxidized CNT fibers (positive electrode)/OMC ^3^/CNT(negative electrode)	CMC ^4^/Na_2_SO_4_	21.7 F g^−1^ (23.4 F cm^−3^)	11.3 mWh cm^−3^ ~2.1 W·cm^−^^3^	85% (10,000 cycles)	[56]

^1^ Multi-walled carbon nanotube (MWNT); ^2^ Dimethyl sulfoxide (DMSO); ^3^ Ordered microporous carbon (OMC); ^4^ Carboxymethyl cellulose (CMC).

**Table 2 nanomaterials-11-00003-t002:** Ionic conductivity results of Chit hydrogel with and without two different percentages of NaCl salt.

Sample Name	Conductivity (mS cm^−1^)
Chit wet gel	0.38 ± 0.04
Chit-NaCl 0.5% w v^−1^	11.4 ± 0.2
Chit-NaCl 1% w v^−1^	15.1 ± 0.1
Chit-4% w v^−1^ NaOH	1.12 ± 0.01

**Table 3 nanomaterials-11-00003-t003:** Mechanical properties of coaxial and triaxial fibers.

Sample Name	Young’s Modulus (GPa)	Ultimate Stress (MPa)	Ultimate Strain (%)
PEDOT:PSS fiber [61]	3.3 ± 0.4	125.0 ± 7.1	15.8 ± 1.2
Chit-PEDOT fiber	3.9 ± 0.1	94.1 ± 3.5	8.6 ± 1.9
Chit-PEDOT (+PEG) fiber	5.2 ± 0.6	160.3 ± 6.8	7.5 ± 0.9
CNT- Chit-PEDOT (+PEG) fiber	4.6 ± 0.3	134.8 ± 3.3	4.5 ± 0.4

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
