# Peer review of "Triaxial Carbon Nanotube/Conducting Polymer Wet-Spun Fibers Supercapacitors for Wearable Electronics"

_nanomaterials, 2020, doi:10.3390/nano11010003_

Round 1

Reviewer 1 Report

This manuscript is a paper on Fibers Supercapacitors composed of Carbon Nanotube/Conducting polymer. It is interesting in terms of the structure and properties of the fiber, but it does not meet expectations as an energy storage device. Therefore, adding the following section is expected to make it possible to publish on Nanomaterials.

  1. At Figure 3, the solution used shows tixotropic properties, which is thought to be an important part in forming the fiber shape. Please provide additional explanation such as Tixotropic index.
  2. In terms of energy density, the electrochemical properties suggested by the authors are not satisfactory compared to general supercapacitors. For accurate understanding, please provide a comparison with other literature using similar structures.
  3. When considering the PEDOT:PSS characteristics, there may be a problem with the cycle characteristics. It is recommended to add cycle characteristics.

Author Response

Dear Editors and Reviewers:

We would like to first thank the editor and reviewers for their feedback. The improved version of the manuscript was renamed to emphasize the main aim and stayed focused on the aim of developing, characterisation and modelling of the textile with recommendations for applications and bulk production for commercialisation of the research. The reviewers’ comments are specifically addressed below, with reference to the point in the manuscript into which feedback has been incorporated. Responds to the reviewer’s comments:

Reviewer #1

This manuscript is a paper on Fibers Supercapacitors composed of Carbon Nanotube/Conducting polymer. It is interesting in terms of the structure and properties of the fiber, but it does not meet expectations as an energy storage device. Therefore, adding the following section is expected to make it possible to publish on Nanomaterials.

  • At Figure 3, the solution used shows thixotropic properties, which is thought to be an important part in forming the fiber shape. Please provide additional explanation such as Thixotropic index.

Response:

Thixotropy is defined as the continuous decrease of viscosity with time when the flow is applied to a colloidal sample that has been previously at rest and the subsequent recovery of viscosity in time when the flow is discontinued.[1] Concentrated chitosan solutions were found to have thixotropic behavior since the apparent viscosity of chitosan decreased with increased mixing time.[2] For the experimental conditions studied, chitosan solution behaved like non-Newtonian shear-thinning fluids (discussed in the manuscript). This can be related to the presence of strong intermolecular hydrogen bonding in chitosan spinning dopes and their tendency to form entanglements. In this study, the changes in solution viscosity were only investigated as a function of shear rate, not time, though. This is mainly because as soon as the chitosan spinning solution is injected into the coagulation bath (termination of shear), there is an instantaneous reaction between the NaOH and -NH3+ groups of chitosan chains in the dope. Therefore, the spinning solution in the rest state is not the initial version of the spinning solution anymore, but a coagulated gel. The dominant mechanism involved in the formation of fibres is known to be the coagulation (neutralization of the protonated amino groups) described previously by Fick’s second law, with the kinetics being controlled by NaOH transport toward the gelification/ solidification zone.[3]

  • In terms of energy density, the electrochemical properties suggested by the authors are not satisfactory compared to general supercapacitors. For accurate understanding, please provide a comparison with other literature using similar structures.

Response:

Based on a survey of the available literature, hybrid fiber-based supercapacitors were reported which indicated either better or weaker electrochemical performance compared to the results of this research. A new paragraph is added in the introduction (page 2) followed by a summary table, to discuss, compare and summarise the performance of previously described structures similar to ours (see below)

‘Researchers have investigated several strategies for the preparation of similar multi-ply fiber SCs.[4–15] Lee et al. described a biscrolled yarn-like SC based on CNT/ PEDOT: PSS nanocomposite and a Pt wire with gel H2SO4/PVA polymer electrolyte, which indicated a specific capacitance of 0.46 mF cm-1 and excellent cycling performance under different mechanical modes. [11] In 2014, a two-ply gamma-irradiated CNT (IR-CNT)/PEDOT: PSS yarn SC was reported using PVA-H3PO4 gel electrolyte, resulted in maximum capacitance of 18.5 F g-1 with no significant reduction in capacitance after 600 charge-discharge cycles. A multi-step wet-spinning process was lately employed to produce ternary CNT/MnO2/PEDOT: PSS composite fiber SCs.[5] The assembled SC device exhibited a high specific capacitance of 51.3 F g−1 and good cycling stability of ~84% capacitance retention after 1000 cycles. A summary of reported hybrid similar fiber SCs is presented in Table 1.

Table 1. Summary Performance of reported similar fiber SCs

Device configuration

Electrolyte

Capacitance

Energy density

Power density

Cycle life

Ref.

IR-CNT/PEDOT: PSS

PVA/H3PO4

18.5 F g-1

91% (600 cycles)

[11]

MWNT/PEDOT: PSS

PVA/ H2SO4

0.46 mF cm-1

1.4 mWh cm-3

40 W cm-3

92% (10000 cycles)

[9]

PEDOT-coated polyester fabric

Na2SO4 solution

0.64 F cm −2 and 5.12 F cm −3

PEDOT:PSS fiber

PVA/H3PO4

119 mF cm-2

4.13 µW h cm-2 250 µW cm-2

[4]

Carbon cloth/PEDOT:PS S

PVA/H3PO4

73 F g-1 at 10 mV s-1

2.6 W h kg-1 2880 W kg-1

~100% (2000 cycles) ~95% under bending

[8]

DMSO doped PEDOT:PSS coated onto cellulose/polyester cloth

Sweat (artificial and human)

8.94 F g-1 (10 mF cm-2 ) at 1mV s-1

1.36 Wh kg-1 (1.63 µWh cm-2 ) at 1.31 V

329.70 W kg-1 (0.40 mW cm-2 ) at 1.31 V

87 – 45%

(1000-5000 cycles)

[15]

MnO2/ PEDOT: PSS/ oxidized CNT fibers

H2SO4 aqueous solution

278.6 mF cm-2

125.37 μWh cm-2

92% (3000 cycles)

[16]

Carbon/MnO2/PEDOT:PSS composite fibres

PVA/H3PO4

51.3 F g-1

84.2% (1000 cycles)

[5]

MnO2/ PEDOT: PSS/ oxidized CNT fibers (positive electrode)/ ordered microporous carbon (OMC)/CNT( negative electrode)

Carboxymethyl cellulose sodium (CMC)/ Na2SO4

21.7 F g−1

(23.4 F cm−3)

11.3 mWh cm−3

∼2.1 W·cm−3

85% (10000 cycles)

[17]

  • When considering the PEDOT:PSS characteristics, there may be a problem with the cycle characteristics. It is recommended to add cycle characteristics.

Response:

There was no noticeable reduction in capacitance after 50 charge-discharge cycles even though herein, we have not assessed the performance at a higher number of cycles. However, further investigation into the earlier works (summarised in Table 1, highlighted in the test in Page 2-3) has indicated good cycle stabilities for fiber supercapacitors based on PEDOT: PSS.

Reviewer 2 Report

Foroughi and coworkers present a study polymer-CNT based supercapacitors.

They provide a clear description of the melt-spinning and CNT coating process, and extensive characterization of the mechanical properties and morphology of the uncoated and coated fibers (note that the 4-panel image in Fig. 5 seems to have been repeated 4 times). All of this in preparation to present the supercapacitor properties in Fig. 7, as cyclic voltammetry. They claim large specific capacity and power density - it would be useful to put these values in context by comparison to other systems.

The study is well planned, executed and described, but I'm not sure that it fits well in the selected journal, Nanomaterials. Other mdpi journals would probably better matches, like Materials, Coatings or Physics. Most of the study is dedicated to the properties of the PEDOT:PSS/chitosan coaxial spiun fibers, its preparation and properties. This is not very "nano". Much less attention is devoted to the carbon nanotubes.

Author Response

Dear Editors and Reviewers:

We would like to first thank the editor and reviewers for their feedback. The improved version of the manuscript was renamed to emphasize the main aim and stayed focused on the aim of developing, characterisation and modelling of the textile with recommendations for applications and bulk production for commercialisation of the research. The reviewers’ comments are specifically addressed below, with reference to the point in the manuscript into which feedback has been incorporated. Responds to the reviewer’s comments:

  • They provide a clear description of the melt-spinning and CNT coating process, and extensive characterization of the mechanical properties and morphology of the uncoated and coated fibers (note that the 4-panel image in Fig. 5 seems to have been repeated 4 times). All of this in preparation to present the supercapacitor properties in Fig. 7, as cyclic voltammetry. They claim large specific capacity and power density - it would be useful to put these values in context by comparison to other systems.

Response:

The error has been resolved regarding the reappearance of Figure 5 in the pdf text (highlighted in Page 8).

A comparison with previously reported structures is given in the introduction (page 2) followed by a summary table.

  • The study is well planned, executed and described, but I'm not sure that it fits well in the selected journal, Nanomaterials. Other mdpi journals would probably better matches, like Materials, Coatings or Physics. Most of the study is dedicated to the properties of the PEDOT:PSS/chitosan coaxial spun fibers, its preparation, and properties. This is not very "nano". Much less attention is devoted to the carbon nanotubes.

Response:

The authors appreciate the reviewer for his valuable comments. This work is based on nanostructured conducting polymer and carbon nanotube as demonstrated into the manuscript.